# Measurement of correlated charge noise in superconducting qubits at an underground facility

G. Bratrud [1,2], S. Lewis[1,3], K. Anyang[1,4], A. Colón Cesaní[2], T. Dyson[5,6,7], H. Magoon[1,5,6,7,8], D. Sabhari[2], G. Spahn[1,9], G. Wagner[1], R. Gualtieri[2], N. A. Kurinsky[1,6,7], R. Linehan[1], R. McDermott[9], S. Sussman[1], D. J. Temples [1], S. Uemura [1], C. Bathurst[10], G. Cancelo[1], R. Chen[2], A. Chou [1], I. Hernandez[1,4], M. Hollister[1], L. Hsu[1], C. James[1], K. Kennard[2], R. Khatiwada[1,4], P. Lukens[1], V. Novati[2,11], N. Raha[2], S. Ray[2], R. Ren[2,12], A. Rodriguez[2], B. Schmidt[2,13], K. Stifter[1,6,7], J. Yu[4], D. Baxter [1,2], E. Figueroa-Feliciano [1,2] & D. Bowring [1] ✉

The charge environment of superconducting qubits may be studied through the introduction of controlled, quantified amounts of ionizing radiation. We measure space- and time-correlated charge jumps on a four-qubit device, operating 107 meters below the Earth's surface in a low-radiation, cryogenic facility designed for the characterization of low-threshold particle detectors. The rock overburden of this facility reduces the cosmic ray muon flux by over 99% compared to laboratories at sea level. Combined with $4\pi$ coverage of a movable lead shield, this facility enables quantifiable control over the flux of ionizing radiation on the qubit device. Long-time-series charge tomography measurements on these weakly charge-sensitive qubits capture discontinuous jumps in the induced charge on the qubit islands, corresponding to the interaction of ionizing radiation with the qubit substrate. The rate of these charge jumps scales with the flux of ionizing radiation on the qubit package, as characterized by a series of independent measurements on another energy-resolving detector operating simultaneously in the same cryostat with the qubits. Using lead shielding, we achieve a minimum charge jump rate of $0.19^{+0.04}_{-0.03}$ mHz, almost an order of magnitude lower than that measured in surface tests, but a factor of roughly seven higher than expected based on reduction of ambient gammas alone. We operate four qubits for over 22 consecutive hours with zero correlated charge jumps at length scales above three millimeters.

A growing body of evidence indicates that ionizing radiation affects the performance of superconducting qubits. Correlation has been measured between the flux of ionizing radiation and the qubit energy relaxation rate ($1/T_1$)[1] and, separately, fluxon lifetime[2]. Ionizing events in a chip substrate cause simultaneous errors in multi-qubit processors[3], as well as

other correlated phenomena in planar superconducting devices[4,5]. Furthermore, the presence of correlated qubit errors, and the rate at which they occur in unshielded laboratories, can interfere with the efficacy of error-correcting surface codes[6–8]. In parallel, researchers have observed charge and parity jumps, correlated in time and space, caused by

**Fig. 1 | Ramsey tomography detects qubit charge jumps. a** Example of a single charge tomography scan from one qubit, illustrating the relationship between the qubit's excited state probability $P_1$ and applied offset charge. Each gray point is the average of 200 measurements. The purple band shows a fit to the template, with the width representing standard error across template samples. **b** A different tomography scan consisting of two charge jumps, with the shaded lines corresponding to the new best template fit after each jump. The green (orange) shaded template corresponds to a charge jump with $\Delta q = 0.13e$ $(0.50e) \pm 0.03e$ relative to the blue template, where $e$ is the elementary charge.

environmental gammas and cosmic rays interacting with the qubit substrate[9–11]. Ionizing radiation has also been shown to "scramble" the spectrum of two-level system populations in superconducting qubits[12]. Moreover, operating quantum devices in an underground facility, shielded from cosmic rays, has been shown to reduce the rate of quasiparticle burst events induced by ionizing interactions in device substrates[13]. A common thread in all these studies is the sudden change in the charge environment of the qubit, leading to measurable changes in device performance.

Here we present experimental results intended to probe the charge environment of superconducting qubits in the presence of controlled and quantified amounts of ionizing radiation. Ionizing events produce electron-hole pairs in the chip substrate. The charge environment near the qubit can be altered by such events if some of the electron-hole pair population becomes trapped in the substrate near it. Electrons and holes can become trapped in sub-gap states (i.e., vacancies, defect sites, etc., with electron trapping potential between the valence and conduction bands of the material). These sub-gap sites are expected and are seen even in high-purity silicon[14,15]. The energy difference between these sites and the conduction band can be anything between zero and the gap, which for silicon is around 1 eV. At temperatures near 10 mK, thermal excitations are on the order of 1 μeV. Random thermal excitations will therefore rarely be able to release these charges into the conduction band. Subsequent high-energy events can impart energy to these

trapped charges or produce new ones, creating a dynamic field environment around the qubit.

With the ultimate goal of understanding and modeling these dynamics[16], mildly charge-sensitive qubits are an ideal platform[17,18]. We operate these qubits as electrometers, each with a sensing area for electric fields in the substrate of hundreds of square microns[10]. The ratio $E_J/E_C = 24$ characterizes the charge sensitivity of the qubits in relation to a standard transmon, with Josephson energy $E_J$ and charging energy $E_C$.

## Results

### Controlling the radiation environment of superconducting qubits

The qubit chip was relocated from the Earth's surface at Madison, WI to the Northwestern EXperimental Underground Site (NEXUS) at Fermilab in Batavia, IL. Previously, correlated jumps in offset charge, associated with gamma ray and cosmic ray impacts, were observed in this qubit array. Underground, over 99% of cosmic ray muons are shielded by the rock overburden, creating an environment in which the qubit response to gamma radiation can be studied in isolation. The muon flux underground is ~7 cm$^{-2}$ day$^{-1}$. A movable lead shield provides $4\pi$ coverage to reduce the ambient gamma background by over 99%. We vary the flux of gamma rays incident on the qubit chip using this shield and measure the rate and magnitude of ensuing charge burst events via Ramsey tomography. Other energy-resolving detectors operating simultaneously in NEXUS are used to calibrate the flux and spectrum of ionizing radiation. Additional facility and experimental hardware details are discussed in the Supplementary Information, Section A. This study is in some sense complementary to that of Ref. 11, in which scintillation detectors provide coincidence information between cosmic ray muon events and qubit errors.

For different datasets during this run period, we vary the flux of gammas incident on the qubit chip by opening and closing the lead shield, and we measure the rate of discontinuous changes in offset charge ("charge jumps"), both for individual qubits and for the correlated event rate across pairs of qubits. The work presented here consists of two datasets corresponding to the lead shield being open (ambient gamma flux) and closed (minimal gamma flux). The integrated measurement times in the "Shield Open" (S.O.) and "Shield Closed" (S.C.) configurations are 23.949 and 22.075 h, respectively.

### Response of qubit charge environment to ionizing radiation

A Ramsey sequence ($X/2 - Idle - X/2$) is applied to each qubit, as described in Refs. 9,10. During the idle period $t_{idle}$ of this sequence, the state vector phase $\phi$ evolves as a function of the dimensionless offset charge $n_g$ present on the qubit island,

$$\phi(n_g) = \Delta f_{01} t_{idle} \cos(2\pi n_g), \qquad (1)$$

for charge dispersion $\Delta f_{01}$. Here, $n_g$ is the sum of the charge applied via the bias lines and the intrinsic offset charge. This pulse sequence maps gate charge (modulo $1e$) onto the excited state probability $P_1$ of the qubit. This mapping is designed to be insensitive to the parity of the qubit island, given that the quasiparticle tunneling rate in this device is orders of magnitude faster than the measurement cadence. A discontinuous change in $n_g$, as from an induced electric field following a charge burst event, causes a discontinuity in $P_1$, as shown in Fig. 1. These discontinuities are recorded as charge jumps with a magnitude $\Delta q$. Scanning the charge bias on each qubit across a range of voltages allows for a calibration of $n_g$ values and an extraction of $P_1$ that is less sensitive to charge noise. See Supplementary Information, Section E for further details.

In principle, $n_g$ can be analytically determined from the excited state population $P_1$[9]. However, when multiple burst events occur in the

qubit substrate during a single tomographic scan, and/or when other incoherent noise is present in the system, fitting against this functional form is not an efficient method of detecting charge jumps. Instead, we perform a rolling $\chi^2$ minimization to fit each tomographic scan against a template averaged from ~20 jump-free scans, with the "phase" of $P_1$ floating on $n_g \in (-0.5e, +0.5e)$. Examples of a jumpless scan and a scan with two detected jumps are shown in Fig. 1.

To quantify the efficiency of this method, the equivalent of 400 h of Ramsey tomography scans were simulated for each qubit, and convolved with a Gaussian noise spectrum according to the measured noise in each qubit. Charge jumps are simulated by injecting $P_1$ discontinuities into this dataset at varying intervals and with varying sizes. In all four qubits, this method finds an efficiency of >70% for identifying jumps of magnitude $0.1e \le |\Delta q| \le 0.5e$, with larger values of $|\Delta q|$ aliased down to the measurement interval. The raw extracted rates for each dataset are then divided by this efficiency to obtain the efficiency-corrected rates, which represent the rates of charge jumps in the qubits, agnostic to our analysis methodology. Charge burst rates are extracted for single qubits and for all qubit pairs. The resulting efficiency-corrected rates above a charge jump threshold of $|\Delta q| \ge 0.1e$ for each qubit are displayed in Table 1. The corresponding time-correlated rates across qubit pairs are displayed in Table 2 and the relative jump sizes $|\Delta q|$ of these pairs are shown in Fig. 2. A pair of charge jumps is considered time-correlated if they occur within 44 s of each other, as limited by our data-acquisition methodology. See Supplementary Information, Section E for details.

We compare the rate of qubit charge jumps against the flux of gamma rays present during the collection of S.O. and S.C. datasets. These fluxes were measured by a $Li_2MoO_4$ (LMO) crystal instrumented with a Transition Edge Sensor as a thermistor[19] and located 18.7 cm from the qubit chip in the DR (see Supplementary Information, Sections A and F). We thus perform a direct measurement of the ratio between S.C. and S.O. gamma fluxes. Based on LMO measurements, we determine that the gamma flux in the S.O. case should be a factor of $20 \pm 1$ larger than in the S.C. case, for gamma energies above 150 keV.

### Relating gamma flux to charge burst rates

Averaged over all four qubits, the charge jump rates that we measure for the S.O. and S.C. data are 0.51 and 0.19 mHz, respectively, as shown in Table 1. Therefore, closing the shield reduces the rate of qubit charge bursts by only a factor of 2.7. This rate is over seven times less than expected based on the factor of 20 reduction in gamma flux measured in the LMO detector. A possible explanation for this discrepancy is that in our lowest-background S.C. configuration, we are sensitive to an excess source of charge bursts that is not dominated by the external gamma flux. Potential sources of radiation inside the fridge that might significantly impact the qubit package but not the LMO detector warrant follow up study and assay[20,21]. The expected muon flux through the qubit (for both configurations) is ~0.08 mHz cm$^{-2}$ – too low to explain the observed rates. The NuMI muon neutrino beam[22] was not active during data collection for this work. Neutrino interactions in matter[23], therefore, contributed no additional muon flux during this study.

We infer that the ambient gamma flux does not dominantly contribute to our S.C. dataset, as the measured rate far exceeds the expected rate given the scaling of LMO data. We therefore subtract the S.C. rates from the S.O. data to obtain a reduced burst rate in the S.O. data, induced by ambient gammas, of $0.34^{+0.07}_{-0.06}$ mHz. In subtracting these data to determine the gamma-induced component, we account for an unknown population of bursts by assuming a constant excess rate of jumps present in both the S.O. and S.C. data. We estimate our ambient gamma flux in the S.O. configuration to be approximately five times lower than that measured using a NaI detector in Ref. 10. If we assume, as Ref. 10 does, that the surface burst rate of 1.35 mHz was gamma-dominated, our ambient rate estimate of 0.34 mHz is proportional with that expected reduction in flux.

The distances between each qubit pair are different, per Table 2 and Fig. 2. The smallest separation between qubit pairs (qubits 3 and 4) is 340 μm and the largest separation between pairs (qubits 1 and 4) is 3330 μm. This variable separation between qubit pairs enables some inferences about correlated noise rates. First, it is technically possible that separate burst events could create conditions that mimic correlated charge jumps arising from a single charge burst. In the S.O. data, we measure a correlated charge jump rate in nearby qubit pairs of 0.27 mHz for qubits 1 and 2, and 0.29 mHz for qubits 3 and 4. Given the low single-qubit jump rates in Table 1, the rate at which this stochastic coincidence could occur is orders of magnitude lower than the measured correlated jump rate. Next, this correlated rate is approximately half of the single-qubit rate (to within statistical uncertainty), which is consistent with that observed in Ref. 10 in which the rate is dominated by gamma flux. In the S.O. data, the correlated jump rate across distant qubit pairs is too low to make any statements on burst origin with any statistical significance. The same statistical limitation is true for nearby qubit pairs in the S.C. data. Finally, we are able to eliminate correlated charge noise in charge-sensitive qubits separated by over 3 mm, on timescales nearing one day.

### Discussion

In summary, we present results from a charge-sensitive qubit chip operated in an underground environment with two configurations of our lead shield. In the shield open configuration, we observe a reduction in charge burst events commensurate with the reduction in

### Table 1 | Charge jump rates in individual qubits

|  | Shield open | Shield closed | Units |
|---|---|---|---|
| Livetime | 23.949 | 22.075 | hours |
| Q1 rate | $0.42^{+0.09}_{-0.08}$ | $0.20^{+0.07}_{-0.05}$ | mHz |
| Q2 rate | $0.60^{+0.11}_{-0.09}$ | $0.19^{+0.07}_{-0.05}$ | mHz |
| Q3 rate | $0.52^{+0.10}_{-0.08}$ | $0.19^{+0.07}_{-0.05}$ | mHz |
| Q4 rate | $0.51^{+0.11}_{-0.09}$ | $0.16^{+0.07}_{-0.05}$ | mHz |
| Average rate | $0.51^{+0.05}_{-0.04}$ | $0.19^{+0.04}_{-0.03}$ | mHz |
| Corrected γ rate | $0.34^{+0.07}_{-0.06}$ | $0.02^{+0.06}_{-0.05}$ | mHz |
| Calculated excess rate | $0.17^{+0.04}_{-0.03}$ | | mHz |

Efficiency-corrected rates (mHz) with magnitude $0.1e \le |\Delta q| \le 0.5e$. Statistical errors are shown; systematic errors are an order of magnitude smaller. Given the apparent non-dependence on external gamma flux for the Shield Closed (S.C.) data, we subtract this from the Shield Open (S.O.) data to find the rate associated only with gamma impacts, as well as the excess jump rate not associated with external gammas. "Livetime" here refers to the total time interval over which data was continuously collected.

### Table 2 | Correlated charge jump rates in qubit pairs

|  | Q1-Q2 | Q3-Q4 | Q1-Q3 | Q1-Q4 | Q2-Q3 | Q2-Q4 | Units |
|---|---|---|---|---|---|---|---|
| Separation | 640 | 340 | 3195 | 3330 | 3180 | 3240 | μm |
| Shield open | $0.27^{+0.09}_{-0.07}$ | $0.29^{+0.09}_{-0.07}$ | $0.03^{+0.04}_{-0.02}$ | $0.08^{+0.06}_{-0.04}$ | $0.05^{+0.05}_{-0.03}$ | $0.08^{+0.06}_{-0.04}$ | mHz |
| Shield closed | $0.10^{+0.07}_{-0.04}$ | $0.04^{+0.05}_{-0.03}$ | < 0.03 | < 0.04 | < 0.03 | < 0.04 | mHz |

Efficiency-corrected rates (mHz) with magnitude $0.1e \le |\Delta q| \le 0.5e$ in each qubit pair. Statistical errors are provided; systematic errors are an order of magnitude smaller. The separation distances of each qubit pair are provided for reference (see Fig. 2). We measure zero correlated charge jumps for qubits separated by over 3 mm, over 22 h of continuous data taking.

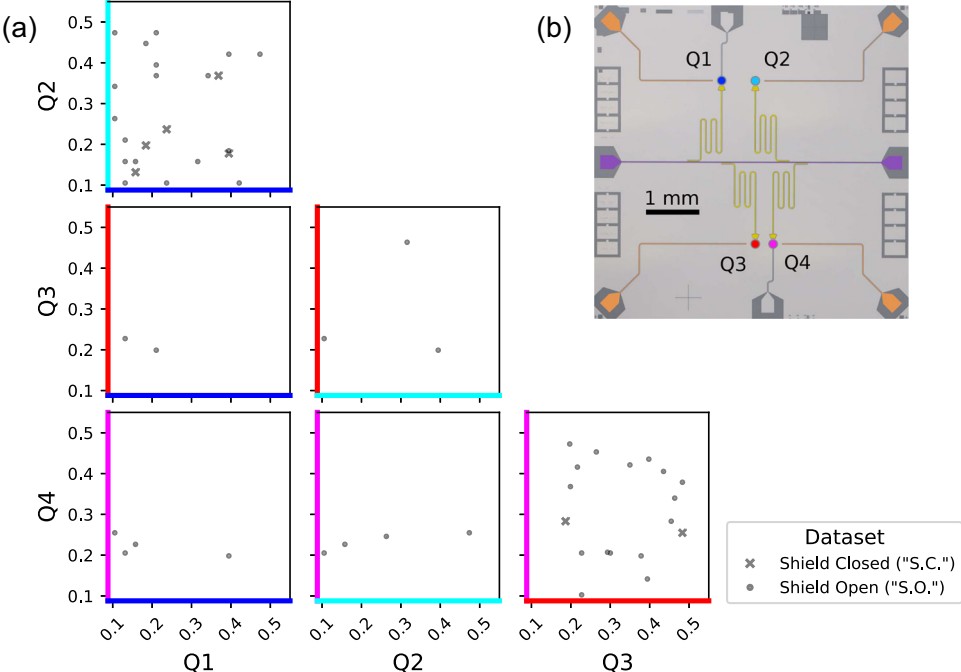

**Fig. 2 | Correlated charge jump magnitudes between qubit pairs depend on distance and shielding. a** Jump magnitudes $0.1e \leq |\Delta q| \leq 0.5e$, in units of electric charge $e$, for all qubit pairs. S.C. data is shown as x's and S.O. data is shown as dots. The raw data retains sign information, but only magnitude $|\Delta q|$ is presented here for visual clarity. Non-correlated events are omitted, also for visual clarity. **b** Micrograph of the qubit chip, annotated with false colors to match the plot axes.

ambient gamma flux relative to the measurements in Ref. 10. Furthermore, in our low-background shield closed dataset, we observe an excess of charge bursts that appears inconsistent with both the expected muon rates and the ambient gamma flux. The next steps will be to investigate the origin of these excess charge bursts. Candidates for this origin include: trapped charge in the substrate that relaxes on long timescales or that is released by IR radiation; secondaries from cosmogenic interactions with the DR materials[20]; and/or an anomalous radiation source inside the dilution refrigerator, very close to the qubit chip. Further study of these events—particularly in low-background environments such as NEXUS—is required to better understand the mechanisms of charge bursts and other errors and their impact on qubit performance. Despite this unexplained excess, our lowest background dataset is free of correlated charge jumps at length scales above 3 mm, during approximately one day of continuous operation. An understanding of the effects of ionizing radiation on the performance of superconducting qubits is critical to the development of these devices for use as particle detectors for fundamental physics[16,24].

## Data availability

The data that support the findings of this study are available from the corresponding authors upon reasonable request and pursuant to US Department of Energy policy.

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

## Acknowledgements

The authors would like to thank Steve Hahn and Cindy Joe for their management of the MINOS underground facility. The authors would like to thank Adam Anderson for providing the QICK board[25] used for these measurements. The TWPA[26] was provided by MIT-Lincoln Laboratory and IARPA. Our use of these devices is discussed in the Supplementary Information, Section B. This manuscript has been authored by Fermi Research Alliance, LLC under Contract No. DE-AC02-07CH11359 with the U.S. Department of Energy, Office of Science, Office of High Energy Physics. This work was supported by the U.S. Department of Energy, Office of Science, National Quantum Information Science Research Centers, Quantum Science Center, and the U.S. Department of Energy, Office of Science, High-Energy Physics Program Office. This work was supported in part by the U.S. Department of Energy, Office of Science, Office of Workforce Development for Teachers and Scientists (WDTS) under the Science Undergraduate Laboratory Internships Program (SULI). This material is based upon work supported by the National Science Foundation Graduate Research Fellowship Program under Grant No. DGE-2234667. Any opinions, findings, and conclusions or recommendations expressed in this material are those of the authors and do not necessarily reflect the views of the National Science Foundation.

## Author contributions

G.B. led the analysis of the results included in this paper, with help from A.C.C. and G.W. S.L. led the RF upgrade of the NEXUS fridge and installation of the qubit payload, with help from D.Bo., T.D., H.M., G.S., N.K., V.N., B.S., and J.Y. G.B., S.L., K.A., R.L., and H.M. contributed to the data-taking scripts used to make this measurement. D.S. led the analysis of the LMO detector data, with help from R.G., R.C., V.N., and B.S. R.G., R.L., S.S., D.J.T., and K.S. provided crucial input on qubit operation. I.A., S.R., and J.Y. provided input through parallel analyses outside of the scope of this paper. S.U. and G.C. provided support with the QICK board and firmware used to take this data. V.N., B.S., and D.J.T. led the operation and maintenance of the NEXUS facility, with help from D.Ba., D.Bo., C.B., R.C., E.F.F., R.G., M.H., C.J., K.K., P.L., N.R., R.R., A.R., and L.H. D.Ba., D.Bo., E.F.F., N.K., and R.M. provided guidance in the scoping and presentation of this work. D.Ba., D.Bo., G.C., A.C., E.F.F., L.H., and R.K. provided leadership of the local quantum group at Fermilab. R.M. provided the qubit chip used in this measurement. D.Bo. coordinated and led this effort as part of his DOE Early Career Award. All authors provided feedback and thoughtful discussions throughout the development of this work.

## Competing interests

The authors declare no competing interests.

## Additional information

[1]Fermi National Accelerator Laboratory, Batavia, IL, USA. [2]Department of Physics & Astronomy, Northwestern University, Evanston, IL, USA. [3]Department of Physics and Astronomy, Wellesley College, Wellesley, MA, USA. [4]Department of Physics, Illinois Institute of Technology, Chicago, IL, USA. [5]Department of Physics, Stanford University, Stanford, CA, USA. [6]Kavli Institute for Particle Astrophysics and Cosmology, Stanford University, Stanford, CA, USA. [7]SLAC National Accelerator Laboratory, Menlo Park, CA, USA. [8]Department of Physics & Astronomy, Tufts University, Medford, MA, USA. [9]Department of Physics, University of Wisconsin-Madison, Madison, WI, USA. [10]Department of Physics, University of Florida, Gainesville, FL, USA. [11]Present address: LPSC, Centre National de la Recherche Scientifique, Université Grenoble Alpes, Grenoble, France. [12]Present address: Department of Physics, University of Toronto, Toronto, ON, Canada. [13]Present address: IRFU, Alternative Energies and Atomic Energy Commission, Université Paris-Saclay, Gif-sur-Yvette, France.
✉e-mail: dbowring@fnal.gov

