## [Transparent Peer Review file · Nature Communications]

Measurement of Correlated Charge Noise in Superconducting Qubits at an Underground Facility

Corresponding Author: Dr Daniel Bowring

Version 0:

Reviewer comments:

Reviewer #1

(Remarks to the Author)

The authors report the measurement of space- and time-correlated charge jumps in a device comprising four superconducting qubits and operating in an underground facility exposed to low radiation. Measurements performed on the same device above ground were previously reported in [Nature 594, 369 (2021) - Ref. 7]. The authors find a sizeable reduction of the charge jumps compared to Ref. 7, consistent with the reduction in the ambient gamma flux. Controlling the gamma flux on the device by opening and closing a lead shield, the authors demonstrate that the rate of charge jumps is not dominated by ambient gamma radiation in the closed-shield configuration.

While the results of the manuscript are potentially interesting and may trigger further research on this topic, we find the manuscript not suitable for publication in Nature Communication in the present form for the following reasons:

- 1) The clarity and accessibility of the manuscript should be improved. In the opening paragraph, it is stated that ionizing radiation affects the qubit's coherence and that correlated errors are relevant for quantum error correction. However, the connection between charge noise and quantum computing is not satisfactorily discussed; charge jumps are not necessarily a source of error, especially for transmons, which are almost insensitive to charge noise. Similarly, it is written that transmons can operate as sensitive electrometers with a sensing area of hundreds of square microns (with no reference). However, it is not explained why correlated errors can be detected over hundreds of microns distance (which is due to charge diffusion). Some of this essential information can be understood only after carefully reading some references, such as Refs. 6 and 7. At the same time, there are two extended paragraphs, the NEXUS paragraph on page 2 and the paragraph commenting on Ba and Cs sources on page 4, which are not essential for the understanding of the paper and should be moved to the supplementary information.
- 2) The reference list is somewhat limited, and references are sometimes improperly cited. When mentioning correlated errors and surface codes, a relevant reference could be [Phys. Rev. Lett. 129, 240502 (2022)]. When commenting on charge-sensitive transmons (page 1), it would be appropriate to include some relevant works on the topic, such as [Nat Commun 4, 1913 (2013)] and [Phys. Rev. B 77, 180502(R) (2008)]. Just before Eq. (1), it would be useful to explicitly mention Ref. [6] for details on the Ramsey sequence, possibly explaining that it is also insensitive to quasiparticle-related parity-switching events. Regarding improper references, Ref. [2] shows a correlation between reduced radioactivity and fluxon lifetime rather than energy relaxation rate. Similarly, the timescale in Refs. [3] and [11] for recovery after events ranges from milliseconds up to fractions of seconds, and not the hours or days written in the text (see second paragraph on page 1), and we couldn't find mention of time scales in Ref. [10].
- 3) Overall, the significance of the results is somewhat unclear. Moving the device into the underground facility decreases the rate of charge jumps by roughly an order of magnitude. However (see also point 1), it is not clear to what extent this reduction is significant for charge-insensitive devices such as transmon, for other solid-state devices, or for quantum error correction. Generally, we find that some of the claims regarding "first measurement" or "first results" for charge events in superconducting qubits should be avoided; see, for instance, the measurements for resonators in [Nature Commun 12, 2733 (2021)]. More precisely, the statement "we are able to, for the first time, eliminate correlated charge noise in charge-sensitive qubits separated by over 3 mm, on timescales nearing one day" seems incorrect, given the results reported in [Nat Commun 13, 7196 (2022)], where for floating transmons measured above ground no simultaneous jumps between qubits with 1.3 mm separation were observed in a 40 hours operation time (note that the single qubit rate was even lower than in this manuscript).

0.007–0.07 mHz).

For completeness, we also point out a few minor typos/inaccuracies:

-At the beginning of page 4, it is stated: “A pair of charge jumps is considered time-correlated if they occur within 44 s of each other.” How is this specific time determined? The Supplementary Material (section D) reports that a full tomographic scan takes 355 seconds. Also in that section, how is the passive initialization achieved (that is, what are the T_1 and waiting times)?

-Clarify how the “Efficiency corrected rates” (see caption Tables 1 and 2) are computed from the raw extracted rates.

- In the first paragraph, replace “charge and parity errors” “charge and parity jumps” to avoid possible confusion with logical errors in quantum computing applications

-On notation: the symbols “e”, “ E_J ”, and “ E_C ” are not defined in the text. Also, ng should be defined as the “dimensionless charge”.

-“complimentary”-> “complementary” at the end of the first paragraph on page 2.

-“expermental”-> “experimental” in the caption of Fig. S1

- in Table S1, the precision with which qubit frequencies f_{01} (and possibly resonator frequencies) are reported is inconsistent with the fact that frequency dispersions of few MHz are measured

- in Supplemental Material section C, more appropriate references about the effects of quasiparticles on T_1 could be given, see for instance [SciPost Lect Notes 31 (2021)] and/or references there; Ref. [3] does not report direct measurements of T_1 , but rather infers an effective T_1 from detected simultaneous errors in multiple qubits

Reviewer #2

(Remarks to the Author)

Reviewer #3

(Remarks to the Author)

The authors measure space- and time-correlated charge jumps on a four-qubit device, operating 107 meters underground in a low-radiation cryogenic facility. This setup, which offers a 99% reduction in cosmic ray muon flux due to its location and is supplemented with a movable lead shield, allows control of the ionizing radiation flux impinging on the device. The measured rate of charge jumps scales with ionizing radiation flux, verified by independent measurements on a dedicated detector in the same cryostat. Using the lead shield, the authors report a minimum charge jump rate of 0.19 mHz, significantly lower than surface tests but higher than expected from ambient gamma reduction alone. The highlight result is the demonstration of over 22 hours of operation with no correlated charge jumps between qubits spaced three millimeters or further (closer spaced qubits did show correlated events).

The text is proficiently written, concise, and supported by clear and convincing figures. The results are of high and immediate relevance for the community, so I recommend publication of the manuscript as is.

My only suggestion is to include in the prior art citations to the work by L. Swenson et al., APL 96, 263511 (2010) and D. Moore et al., APL 100, 232601 (2012) on correlated quasiparticle bursts due to ionizing radiation interactions with the substrate, and to the work of L. Cardani et al., Nat Commun 12, 2733 (2021) for showing for the first time that operating in an underground facility is beneficial for quantum devices.

Reviewer #4

(Remarks to the Author)

The authors report results of charge jumps in charge-sensitive transmons reporting correlations between qubits on a four-qubit chip. The main novelty lies in the fact that the results have been obtained in an underground laboratory where the cosmic muon flux is largely attenuated, which leaves gamma as the dominant source of ionizing radiation. The authors also study the correlations with or without a lead shield that can be moved to surround the cryostat finding reduced correlations with the shield in place. Furthermore, the authors use a transition edge sensor to calibrate gamma radiation at the sample area and to corroborate their findings of reduced correlated jump rates with the lead shield although some unexplained sources of radiation affecting the qubit chip remain.

Overall, I find the work of high technical quality and it has clearly taken a lot of planning and execution from the authors. The manuscript reads well for the most part and can be of interest to a wide readership. To better understand the significance of this work I am hoping my concerns will be addressed. My comments are itemized below in varying degrees of importance.

-I was looking for a take home message from the manuscript but even reading the abstract, intro and conclusions multiple times I could not quite see it. I realize that this measurement was just a beginning to the authors' work. However, I would like for the authors to clarify their message. Is the purpose of this manuscript to introduce new methodology and the authors' new advanced underground measurement setup or the results they have now obtained with it? Are the authors able to comment on what exactly they are hoping to find in their future measurement campaigns with their new setup? At the very end, the authors mention qubits as particle detectors and implications for quantum computing. Can the authors elaborate on these implications and specifically how it relates to their results and future measurements?

- The authors single out gamma radiation refs [17,18] as the remaining contributor to the SC rates. The possibility of stray IR photons is not acknowledged in the manuscript although the existence of IR activated quasiparticles is well known (for example: Appl. Phys. Lett. 99, 113507 (2011)). It seems from supplementary figure S2 that no radiation shielding apart from the mu-metal magnetic shielding nor IR absorbing black coatings have been utilized. Can the authors examine the importance of IR in their results?

-Although unlikely to have any bearing on the IR radiation, I noticed that the drive line attenuation (fig. S2) also seems lower in contrast to Ref. 7, which can be a source of excess thermal noise at the sample and hence cause dephasing during the Ramsey sequence. It is unclear how decoherence during the measurement sequence affects the measured jump rates particularly since the sequence t_{idle} time nor the T_2^* times are specified in the manuscript. Are these events already handled and discarded during the fitting of tomography data?

-(optional) The authors have correctly identified future directions of performing radioactivity assay of the materials and components within proximity to the qubit chip or even that of the chip materials itself. I understand however that may be outside the scope of this work.

-page 3, final paragraph: "above procedure" seems to refer to the simulation of charge jumps but that is clearly not what the authors meant.

-page 3, final paragraph: does the efficiency of >70 % identifying jumps mean that the measured rates are 70 % of the true rates or is this fact reflected by the error bars? It may be necessary to demonstrate the error analysis in the supplement.

-page 4, top: why has the window for correlations been selected to be precisely 44 s? I realize this condition may come from Ref. 7 but I would find it useful to re-iterate the justification here.

-page 4, first paragraph: conclusions are drawn from the LMO dataset but Supplement has not been referenced yet

-page 4, fourth paragraph: can the authors please clarify why ambient gamma flux does not contribute to the SC dataset? If it is due to the observation of 20-fold reduction in gammas observed in the LMO data but a lesser reduction in the topography measurements, can the authors write that explicitly?

-page 5, figure 2 caption: please fix typo in "micgrograph"

-page 9, second paragraph: please fix typo in "collecatation"

-page 9, figure S1 caption: please fix typo in "expermental"

-page 13, final paragraph: I assume "both datasets" here refer to the Ramsey tomography datasets and not the LMO datasets unlike what the sentence implies. Can the authors improve the wording?

Version 1:

Reviewer comments:

Reviewer #1

(Remarks to the Author)

The authors have taken into consideration in an appropriate manner all the referees' comments, with one exception. After they consider the few remarks below, the manuscript can be considered for publication. However, we do not find Nature Communication to be the most appropriate journal: the advances reported in this paper are rather technical, as they mostly show that a new facility can enable more sensitive measurements; therefore, we think that the article should be transferred to a more specialized journal (such as Communication Physics).

Unresolved comment:

One unwarranted claim of priority has not been removed, since the sentence "Finally, we are able to, for the first time, eliminate correlated charge noise in charge-sensitive qubits separated by over 3 mm, on timescales nearing one day." on page 5 has not been revised. As mentioned in our previous report, the results reported in Nat Commun 13, 7196 (2022) (no simultaneous jumps between qubits separated by 1.3 over 40 hours) contradict this claim and should be referred to in the context of this work.

Minor comments:

- Typo: efficiency-correct -> efficiency-corrected in the first paragraph on page 3.

- Ref. 11 should be updated with the published version: Nat Commun 16, 6428 (2025)

- In the second paragraph, beginning of pag.2, it is no longer stated that the measured device is the same as in Ref.10. This information should be provided, since it would clarify the sentence at the beginning of the third paragraph: "The qubit chip was relocated..."

Reviewer #2

(Remarks to the Author)

Reviewer #3

(Remarks to the Author)

I am satisfied with the revision, and I also find the author's replies to the other reviewers satisfactory. I recommend publication.

Reviewer #4

(Remarks to the Author)

My concerns have been adequately addressed. I appreciate the added clarity and technical detail in this new version. I do not have any further comments and recommend the manuscript for publication.

14 July 2025

Daniel Bowring
Staff Scientist

**Emerging Technologies
Directorate**
P.O. Box 500, MS 209
Kirk Road and Pine Street
Batavia, Illinois 60510-5011
USA
Office: 630.840.6704
dbowring@fnal.gov

To the referees of manuscript NCOMMS-24-49029-T:

We thank you for your thoughtful and substantive comments on our manuscript, NCOMMS-24-49029-T, now titled "Measurement of Correlated Charge Noise in Superconducting Qubits at an Underground Facility". We believe we have responded substantially to all comments. All responses are discussed in depth, below, following the order of comments in each referee's response.

Our resubmission includes a version of the manuscript, NEXUS_qubits_highlighted.pdf, in which the referees' suggested changes are highlighted (where applicable). This conforms to Nat. Comms. editorial policy, and may also be useful during review.

We believe this manuscript is stronger as a result of our reviewers' input and we thank them and the editors of Nature Communications for their time and patience.

Thanks very much for your consideration,

Daniel Bowring

Comments from Reviewers 1 & 2 are shown in **bold text**, along with our responses and indications of relevant changes to the manuscript.

The clarity and accessibility of the manuscript should be improved. In the opening paragraph, it is stated that ionizing radiation affects the qubit's coherence and that correlated errors are relevant for quantum error correction. However, the connection between charge noise and quantum computing is not satisfactorily discussed; charge jumps are not necessarily a source of error, especially for transmons, which are almost insensitive to charge noise.

We appreciate the reviewers' point that our opening paragraph did not sufficiently distinguish between decoherence errors and charge bursts. Both effects are clearly important but distinct. We have altered the language of the first paragraph to clarify this distinction.

Similarly, it is written that transmons can operate as sensitive electrometers with a sensing area of hundreds of square microns (with no reference). However, it is not explained why correlated errors can be detected over hundreds of microns distance (which is due to charge diffusion). Some of this essential information can be understood only after carefully reading some references, such as Refs. 6 and 7.

This comment seems to stem from a lack of clarity on our part in the relevant text. We agree that it is not generally true that transmons can operate as electrometers with $O(100)$ square-micron sensing area. However, in this specific case this is accurate, and supported by electrostatic simulations. To clarify, we add a reference to Wilen [10] at the discussion of electrometer area sensitivity. For the specific chip (not transmons, but slightly charge-sensitive qubits) used in that paper and in this manuscript, this is true. The electric field distribution in that chip is simulated in Fig. 1 of that paper. In this chip, correlated charge noise stems from a combination of charge diffusion (as mentioned) and also the radius of sensitivity to charge trapping around the qubit island. Correlated errors are detectable over distances larger than hundreds of microns because of the track length of scattered electrons in the silicon substrate.

At the same time, there are two extended paragraphs, the NEXUS paragraph on page 2 and the paragraph commenting on Ba and Cs sources on page 4, which are not essential for the understanding of the paper and should be moved to the supplementary information.

We have moved the Ba/Cs paragraph to the supplementary material. We agree that this material is not critical for an understanding of the paper results, but felt it was important to explicitly acknowledge data that was collected and *not* used for the analysis in this manuscript. The supplementary material is an appropriate place for this type of discussion. A small amount of original material on NEXUS is essential for the understanding of the paper. That material has been preserved, with the rest of the facility information moved to the supplement.

The reference list is somewhat limited, and references are sometimes improperly cited. When mentioning correlated errors and surface codes, a relevant reference could be [Phys. Rev. Lett. 129, 240502 (2022)]. When commenting on charge-sensitive transmons (page 1), it would be appropriate to include some relevant works on the topic, such as [Nat Commun 4, 1913 (2013)] and [Phys. Rev. B 77, 180502(R) (2008)]. Just before Eq. (1), it would be useful to explicitly mention Ref. [6] for details on the Ramsey sequence, possibly explaining that it is also insensitive to quasiparticle-related parity-switching events. Regarding improper references, Ref. [2] shows a correlation between reduced radioactivity and fluxon lifetime rather than energy relaxation rate.

We have expanded the references per the referee's suggestion. See, for example, Refs. 17 and 18 in the revised manuscript. Language has been added at the beginning of p. 2 to improve clarity on the motivation for choosing charge-sensitive qubits, and for choosing the particular Ramsey sequence that we use.

Similarly, the timescale in Refs. [3] and [11] for recovery after events ranges from milliseconds up to fractions of seconds, and not the hours or days written in the text (see second paragraph on page 1), and we couldn't find mention of time scales in Ref. [10].

We have changed this paragraph to address the referee's points. Indeed timescales are hard to predict. In principle at 10mK $k_B T$ excitations should not be able to release a charge trapped in a shallow sub-gap site that might need of order 0.1-1 eV of energy to be released into the conduction band. We assume that the timescales are dictated by the rate of excitations from gammas or other external factors, but this is very dependent on both the environment and the sub-gap spectrum of the sample. We have removed mentions of time scales and just make the point that we expect these trapped charges to be long-lived and measurable by our devices.

Overall, the significance of the results is somewhat unclear. Moving the device into the underground facility decreases the rate of charge jumps by roughly an order of magnitude. However (see also point 1), it is not clear to what extent this reduction is significant for charge-insensitive devices such as transmons, for other solid-state devices, or for quantum error correction. Generally, we find that some of the claims regarding "first measurement" or "first results" for charge events in superconducting qubits should be avoided; see, for instance, the measurements for resonators in [Nature Commun 12, 2733 (2021)]. More precisely, the statement "we are able to, for the first time, eliminate correlated charge noise in charge-sensitive qubits separated by over 3 mm, on timescales nearing one day" seems incorrect, given the results reported in [Nat Commun 13, 7196 (2022)], where for floating transmons measured above ground no simultaneous jumps between qubits with 1.3 mm separation were observed in a 40 hours operation time (note that the single qubit rate was even lower than in this manuscript 0.007–0.07 mHz).

The focus of this paper, and on other works by this group cited therein, is on understanding the microphysics of ionization events in superconducting devices. We agree with the reviewers' comment that this focus was muddled, given the "first measurement" (and other) language. We have therefore modified the title of the manuscript and some of the language used throughout the paper to help put focus where it is intended: on the microphysics. See, for example, the new material on pages 1-2 detailing the electron/hole dynamics of interest.

At the beginning of page 4, it is stated: "A pair of charge jumps is considered time-correlated if they occur within 44 s of each other." How is this specific time determined? The Supplementary Material (section D) reports that a full tomographic scan takes 355 seconds.

We have added a clarifying sentence to the main text and the following paragraph to the Supplemental Material: "Our time-correlation window of 44 s is limited by our data-acquisition methodology. This time-scale comes from the limitations of time-correlation. Charge jumps in 2+ qubits are considered correlated if they are identified within a window of 10 points of one another in the same tomography scan. Since each point takes 0.98 s for qubits 1-3 and 1.4 s for qubit 4, and since each data point is taken consecutively for each qubit, two neighboring points for the same qubit are about 4.4 s apart in time. We

chose a window of 10 points (44 s) for identifying correlated jumps due to the “time” (i.e. number of points) it takes for an average-size charge jump to be identified through our minimum combined χ^2 methodology. Since the minimum combined χ^2 has to rise above a threshold in order to be identified as a jump this can cause the jump to be identified a few points after it happens in the scan.”

Also in that section, how is the passive initialization achieved (that is, what are the T₁ and waiting times)?

Our T₁s were approximately 10-20 us depending on the qubit, so we selected 200 us as an appropriate initialization time. This is ten times our highest T₁ (~20us). We have added text to Supplement D to clarify this point.

Clarify how the “Efficiency corrected rates” (see caption Tables 1 and 2) are computed from the raw extracted rates.

We have added the following sentence at the top of p. 4: “The raw extracted rates for each dataset are then divided by this efficiency to obtain the efficiency-correct rates, which represent the rates of charge jumps in the qubits, agnostic to our analysis methodology.”

In the first paragraph, replace “charge and parity errors” with “charge and parity jumps” to avoid possible confusion with logical errors in quantum computing applications.

We have done this.

On notation: the symbols “e”, “E_J”, and “E_C” are not defined in the text. Also, ng should be defined as the “dimensionless charge”.

These changes were made near the p. 2 and 3, respectively.

In Table S1, the precision with which qubit frequencies f₀₁ (and possibly resonator frequencies) are reported is inconsistent with the fact that frequency dispersions of few MHz are measured.

Indeed, the precision of these reported values was inconsistent. We have checked against our archived data and corrected this inconsistency in the manuscript.

In Supplemental Material section C, more appropriate references about the effects of quasiparticles on T₁ could be given, see for instance [SciPost Lect Notes 31 (2021)] and/or references there; Ref. [3] does not report direct measurements of T₁, but rather infers an effective T₁ from detected simultaneous errors in multiple qubits.

We have replaced the citation to the McEwen paper with something more focused on QP dynamics. Instead of the reference suggested by the reviewer, we have selected one of the references included therein, which we feel to be more appropriate: <https://doi.org/10.1103/PhysRevB.84.064517>

We thank Reviewer 3 for their summary of our work and the recommendation to publish as-is. Comments from Reviewer 3 are shown in **bold text**, along with our responses and indications of relevant changes to the manuscript.

My only suggestion is to include in the prior art citations to the work by L. Swenson et al., APL 96, 263511 (2010) and D. Moore et al., APL 100, 232601 (2012) on correlated quasiparticle bursts due to ionizing radiation interactions with the substrate, and to the work of L. Cardani et al., Nat Commun 12, 2733 (2021) for showing for the first time that operating in an underground facility is beneficial for quantum devices.

We have added sentences and citations in the introductory paragraph as suggested.

We thank Reviewer 4 for their kind remarks. Comments from Reviewer 4 are shown in **bold text**, along with our responses and indications of relevant changes to the manuscript.

I was looking for a take home message from the manuscript but even reading the abstract, intro and conclusions multiple times I could not quite see it. I realize that this measurement was just a beginning to the authors' work. However, I would like for the authors to clarify their message. Is the purpose of this manuscript to introduce new methodology and the authors' new advanced underground measurement setup or the results they have now obtained with it? Are the authors able to comment on what exactly they are hoping to find in their future measurement campaigns with their new setup? At the very end, the authors mention qubits as particle detectors and implications for quantum computing. Can the authors elaborate on these implications and specifically how it relates to their results and future measurements?

See Reviewer 1 & 2's comments above. We have revised language in the paper to place a stronger emphasis on the intended focus of the paper: the microphysics of ionizing events in superconducting devices. These changes should be evident in pages 1-2, as well as in the concluding paragraphs.

The authors single out gamma radiation refs [17,18] as the remaining contributor to the SC rates. The possibility of stray IR photons is not acknowledged in the manuscript although the existence of IR activated quasiparticles is well known (for example: Appl. Phys. Lett. 99, 113507 (2011)). It seems from supplementary figure S2 that no radiation shielding apart from the mu-metal magnetic shielding nor IR absorbing black coatings have been utilized. Can the authors examine the importance of IR in their results?

We thank the reviewer for pointing out our omission of this possibility. Indeed, IR photons could release electrons trapped in shallow impurity sites close to the conduction band. We have added text to list this as a possibility and will look into it in future work.

Although unlikely to have any bearing on the IR radiation, I noticed that the drive line attenuation (fig. S2) also seems lower in contrast to Ref. 7, which can be a source of excess thermal noise at the sample and hence cause dephasing during the Ramsey sequence. It is unclear how decoherence during the measurement sequence affects the measured jump rates particularly since the sequence t_idle time nor the T₂^{*} times are specified in the manuscript. Are these events already handled and discarded during the fitting of tomography data?

We thank the reviewer for pointing out that we omitted these important parameters from our manuscript. We have added them to the text. The idle time is set as $1/(4\Delta f_{01})$ for each qubit, where Δf_{01} is the charge dispersion, listed in Table S1. For qubit 4, for example, this idle time is 0.07 μ s. We measure a $T_2^* > 1$ μ s. We believe this dephasing time is sufficiently long compared to our pulse timing so that we don't see decoherence during the measurement. The reviewer is correct that our thermal population was high in this result; we have corrected this for subsequent measurements, to be described in subsequent manuscripts.

(optional) The authors have correctly identified future directions of performing radioactivity assay of the materials and components within proximity to the qubit chip or even that of the chip materials itself. I understand however that may be outside the scope of this work.

The reviewer correctly points out that radioactivity in the materials near our qubit package (but far from the LMO) could be the source of the excess background we measure, as we state in the paper, but the process for radio-assaying these materials is out of the scope of this result. This is a subject of future work.

In the final paragraph of p. 3, "above procedure" seems to refer to the simulation of charge jumps but that is clearly not what the authors meant.

The intended meaning of this clause was actually redundant with an earlier part of the paragraph, so this text has been removed.

Does the efficiency of >70 % identifying jumps mean that the measured rates are 70 % of the true rates or is this fact reflected by the error bars? It may be necessary to demonstrate the error analysis in the supplement.

We have added an additional sentence defining what we mean by efficiency-corrected rates. The measured rates are ~70% of the true rates. The error bars include the propagated errors in our understanding of this efficiency.

why has the window for correlations been selected to be precisely 44 s? I realize this condition may come from Ref. 7 but I would find it useful to re-iterate the justification here.

See above response to the same comment from Reviewer 1.

Conclusions are drawn from the LMO dataset but Supplement has not been referenced yet.

We have added a reference to the supplemental material at this first mention of the LMO.

Finally, we thank all reviewers for catching various typos and style issues. These have been corrected.

14 August 2025

Daniel Bowring
Staff Scientist

**Emerging Technologies
Directorate**
P.O. Box 500, MS 209
Kirk Road and Pine Street
Batavia, Illinois 60510-5011
USA
Office: 630.840.6704
dbowring@fnal.gov

To the reviewers of manuscript NCOMMS-24-49029A:

We thank you for your comments on our revised manuscript, “Measurement of Correlated Charge Noise in Superconducting Qubits at an Underground Facility”. This letter describes our responses to these comments.

Reviewer 1 had one unresolved comment: “One unwarranted claim of priority has not been removed, since the sentence “Finally, we are able to, for the first time, eliminate correlated charge noise in charge-sensitive qubits separated by over 3 mm, on timescales nearing one day.” on page 5 has not been revised.” This was an oversight on our part. We had intended to remove all such uses of “first”, or “for the first time”. (c.f. our revised manuscript title.) This version of the manuscript corrects this oversight.

Reviewer 1 also points out several minor issues: a typo, and a reference that required updating from an arxiv entry to a Nature Communications article. These comments have been addressed.

Reviewer 2 is a co-reviewer; their feedback is incorporated into another reviewer’s comments and it is not possible to address their contributions specifically.

We thank Referees 3 and 4 for their recommendation to publish.

Sincerely,

Daniel Bowring